# Pro-Arrhythmic Potential of Accumulated Uremic Toxins Is Mediated via Vulnerability of Action Potential Repolarization

**DOI:** 10.3390/ijms24065373

**Published:** 2023-03-11

**Authors:** Willem B. van Ham, Carlijn M. Cornelissen, Elizaveta Polyakova, Stephanie M. van der Voorn, Merel L. Ligtermoet, Jantine Monshouwer-Kloots, Marc A. Vos, Alexandre Bossu, Eva van Rooij, Marcel A. G. van der Heyden, Toon A. B. van Veen

**Affiliations:** 1Department of Medical Physiology, Division Heart & Lungs, University Medical Center Utrecht, 3584 CX Utrecht, The Netherlands; 2Hubrecht Institute, Royal Netherlands Academy of Arts and Sciences (KNAW), University Medical Center Utrecht, 3584 CX Utrecht, The Netherlands

**Keywords:** cellular electrophysiology, uremic toxins, chronic kidney disease, K_V_11.1, cardiac repolarization

## Abstract

Chronic kidney disease (CKD) is represented by a diminished filtration capacity of the kidneys. End-stage renal disease patients need dialysis treatment to remove waste and toxins from the circulation. However, endogenously produced uremic toxins (UTs) cannot always be filtered during dialysis. UTs are among the CKD-related factors that have been linked to maladaptive and pathophysiological remodeling of the heart. Importantly, 50% of the deaths in dialysis patients are cardiovascular related, with sudden cardiac death predominating. However, the mechanisms responsible remain poorly understood. The current study aimed to assess the vulnerability of action potential repolarization caused by exposure to pre-identified UTs at clinically relevant concentrations. We exposed human induced pluripotent stem cell-derived cardiomyocytes (hiPSC-CMs) and HEK293 chronically (48 h) to the UTs indoxyl sulfate, kynurenine, or kynurenic acid. We used optical and manual electrophysiological techniques to assess action potential duration (APD) in the hiPSC-CMs and recorded I_Kr_ currents in stably transfected HEK293 cells (HEK-hERG). Molecular analysis of K_V_11.1, the ion channel responsible for I_Kr_, was performed to further understand the potential mechanism underlying the effects of the UTs. Chronic exposure to the UTs resulted in significant APD prolongation. Subsequent assessment of the repolarization current I_Kr_, often most sensitive and responsible for APD alterations, showed decreased current densities after chronic exposure to the UTs. This outcome was supported by lowered protein levels of K_V_11.1. Finally, treatment with an activator of the I_Kr_ current, LUF7244, could reverse the APD prolongation, indicating the potential modulation of electrophysiological effects caused by these UTs. This study highlights the pro-arrhythmogenic potential of UTs and reveals a mode of action by which they affect cardiac repolarization.

## 1. Introduction

Cardiorenal syndrome (CRS) is a global health care problem with high morbidity and mortality [1,2,3]. The pathology of this syndrome is characterized by acute or chronic dysfunction of one of these organs, subsequently leading to malfunction of the other. CRS consists of a classification of five subtypes in which the bidirectional interaction between the heart and kidneys is displayed [4,5]. Primary chronic kidney disease (CKD), a gradual loss of function of the kidneys that eventually may lead to end-stage renal disease (ESRD), contributes to reduced cardiac function and/or increased risk for adverse cardiovascular events [6]. Almost half of the deaths in CKD patients are caused by these cardiovascular events, particularly sudden cardiac death (SCD) in ESRD patients [7,8].

Due to the decline in kidney function, filtration of uremic solutes is compromised, leading to increased concentrations of uremic toxins (UTs) and other waste products in the circulation. In addition, the accumulation of UTs also further contributes to progressive damage to the kidneys, thereby worsening CKD [4,9]. ESRD patients mainly rely on dialysis to filter their blood; however, dialysis only mimics the glomerular filtration of the kidneys and not the tubular section mediated by transporters. Protein-bound uremic toxins (PBUTs) are predominantly excreted by tubular secretion due to their strong protein-binding properties [10,11]. This process results in accumulation of these toxins even in dialysis patients since only the free fraction of the UTs can be filtered in conventional dialysis therapies [12].

The high concentrations of UTs in ERSD, in association with the high SCD rates in these patients, suggests an important effect on cardiac electrophysiology that has been scarcely investigated in basic experimental studies [13]. Prolongation of the action potential duration (APD), as a trigger for increased susceptibility of early after depolarizations (EADs), is a known pro-arrhythmic parameter preceding SCD. Studying deviations in cardiac repolarization could prove beneficial in identifying electrophysiologically dangerous UTs, laying the ground for a more specific approach to UT removal. While some experimental studies have shown systemic effects evoking altered calcium handling and sodium currents in models of CKD [14,15,16], the majority of data have predominately indicated effects on potassium currents, as recently reviewed by our group [13]. K_V_11.1, or hERG, comprises one of the most important ion channels involved in repolarization of the action potential. These channels are highly sensitive to blockage by a variety of compounds and drugs, accounting for the high attrition rate during drug development. This sensitivity stresses the need to screen for potential pro-arrhythmogenicity of UTs by focusing on the repolarization current I_Kr_ [17].

The aim of this study was to identify the consequences on cardiac repolarization of UT exposure, using a subset of clinically relevant UTs, including indoxyl sulfate (IS), kynurenine (KYN), and kynurenic acid (KYNA). First, we determined the influence of acute and chronic UT exposure on APD in human induced pluripotent stem cell-derived cardiomyocytes (hiPSC-CMs). Since chronic exposure resulted in APD prolongation, we measured current densities of the repolarization current I_Kr_ using a cell line stably expressing K_V_11.1 (HEK-hERG). At the molecular level we analyzed levels of mRNA and protein of K_V_11.1 after chronic UT exposure. Finally, we performed an intervention strategy using LUF7244, an activator of K_V_11.1 channels, in an attempt to recover repolarization functionality in hiPSC-CMs after chronic UT exposure. In this study, we identified the negative effects of three UTs on cardiac repolarization, while distinguishing the more clinically relevant UTs and concentrations.

## 2. Results

### 2.1. Action Potential Prolongation Caused by Chronic Toxin Exposure

First, using the organic voltage-sensitive fluorescent dye FluoVolt, we measured APD in hiPSC-CMs (Figure 1A). Cardiomyocytes were exposed acutely (5 min) to one of two concentrations of each UT. Dofetilide, a well-known blocker of the cardiac repolarization hERG channel, was used as positive control (Figure 1B). Mean APD in the control group (304 ± 71 ms) remained similar upon exposure to all UT conditions, except for dofetilide, which significantly prolonged repolarization (APD 547 ± 206 ms). After chronic exposure (48 h), the APD remained unchanged in the control group (297 ± 60 ms), while all other conditions significantly prolonged the APD (343 ± 77 and 353 ± 73 ms, 513 ± 95 and 460 ± 59 ms, and 375 ± 100 and 393 ± 112 ms for the low and high concentrations of IS, KYN, and KYNA respectively) (Figure 1C). This finding clearly suggests that chronic, but not acute, exposure to UTs is necessary to trigger a prolongation of repolarization in hiPSC-CMs. Therefore, it translates to the clinical settings, whereby patients are chronically exposed to high concentrations of UTs regardless of dialysis. Interestingly, no concentration-dependent effect was seen with any of the UTs, although the high concentrations of KYN and KYNA far exceeded the pathological concentrations measured in CKD patients. The lowest concentrations applied are in line with those measured in patients.

### 2.2. Decreased I_Kr_ Current Density After Exposure to Clinically Relevant UT Concentrations

Next, to elucidate the underlying mechanism of APD lengthening, we focused on the rapid delayed rectifier potassium repolarization current I_Kr_. Using a voltage clamp in HEK-hERG cells, we measured I_Kr_ currents with or without chronic exposure (48 h) to the UT concentrations, including the clinically relevant concentrations, which are shown as example traces (Figure 2A). The current densities, which are comparable to relevant hERG channel activation during the repolarization, are covered between +20 and +40 mV [18] (Figure 2B). Exposure to both concentrations of IS resulted in a diminished peak current density compared to control measurements (35 and 32 vs. 51 pA/pF for 3 and 30 µM respectively), again implying that no difference was observed between the clinically relevant and the exceeding concentration of IS. KYN only decreased the I_Kr_ current at 30 µM (20 vs. 51 pA/pF), whereas the clinically relevant concentration of 3 µM did not have an effect. Exposure to KYNA also resulted in smaller currents for both concentrations (34 and 21 vs. 52 pA/pF). Although not significant (*p* = 0.08), 10 µM KYNA decreased the I_Kr_ current even further than the more clinically relevant 1 µM. While the hampered hERG channel functionality could explain the prolonged APDs as being caused by the UTs, the APD prolongation caused by 3 µM KYN did not seem to be disturbed in an I_Kr_-dependent manner.

To gain more insight in the molecular events underlying this vulnerability in repolarization, we quantified the protein and mRNA expression of the hERG channel as a potential cause of the diminished I_Kr_ current. Similar to the decreased currents, protein levels were reduced for the UT conditions in a dose-independent manner (Figure 3A,B). IS at a concentration of 30 µM and 10 µM KYNA exposure resulted in a reduction of approximately 70%. hERG protein decreases due to 3 µM KYN were nonsignificant compared to control. mRNA levels of the *KCNH2* gene were not altered in any of the conditions compared to control (Figure 3C), indicating a (post-)translational effect of the UTs that resulted in lower protein levels.

### 2.3. Action Potential Prolongation Can Be Reversed by LUF7244

The reduced expression of functional protein could limit the possibility of recovering from prolonged UT exposure. We therefore attempted to reverse the APD prolongation by treatment with LUF7244, an activator of the K_V_11.1 ion channel [19]. hiPSC-CMs were exposed for 48 h with a high concentration of the three UTs, followed by an additional 24 h of UT exposure either with or without supplementation of 10 µM LUF7244 (Figure 4A,B). UT exposure prolonged the APD for all three toxins, similar to the earlier described experiments (547 ± 139, 522 ± 141, and 601 ± 205 ms for IS, KYNA, and KYNA respectively). Control APDs were approximately 100 ms longer compared to previous measurements (390 ± 92 vs. 297 ± 60 ms), presumably due to known fluctuations between hiPSC-CMs differentiations. Addition of LUF7244 robustly shortened the APD for all groups (177 ± 79, 170 ± 66, 159 ± 57, and 180 ± 78 ms for control, IS, KYN, and KYNA respectively).

## 3. Discussion

In the present study we described the electrophysiological effect of UTs, which accumulate in the bloodstream of CKD patients. Chronic exposure to three clinically relevant UTs resulted in prolongation of the APD of hiPSC-CMs, although no acute effect was observed. These UTs also diminished the I_Kr_ current in HEK-hERG cells, indicating that this current, which is of major importance for adequate repolarization, could be responsible for lengthening of the APD. Additionally, chronic exposure to UTs appeared to lower K_V_11.1 protein levels, an event not caused by alterations in *KCNH2* mRNA expression. Functional restoration of the APD in hiPSC-CMS could be achieved with LUF7244 treatment, indicating that sufficient repolarization current can be rescued to stabilize the action potential in the presence of the three UTs.

The high concentrations of UTs in CKD patients have been linked to inflammation and fibrosis formation in the past [20,21]. However, there have been a limited number of studies that have investigated the potential direct electrophysiological effects of UTs. Previously, exposure to IS and another UT, p-cresyl sulfate, has been shown to decrease repolarization currents in H9c2 rat ventricular cardiomyocytes and computer simulations [22,23]. Additional studies, including in vivo studies, have shown pro-arrhythmic consequences but failed to delve into potential mechanisms of action [13]. With this current study, we have shown that UTs can additionally influence ion channel functionality, resulting in electrophysiological disturbances that might lead to increased susceptibility to arrhythmias in CKD patients. Future in-dept molecular analysis of the mechanisms triggered through exposure to different UTs could provide more specific targets to overcome the functional problems caused by any UT, rather than a solution against all toxins, which would demand a much broader and elaborative approach. While this study, as well as the majority of current electrophysiological research on UTs, only describes the repolarization current I_Kr_, it is possible that other ion currents, such as late sodium, are also affected by the UTs and could be responsible for APD prolongation. This possibility is especially interesting for KYN since it showed a large APD lengthening but no effects on I_Kr_. Future studies should also include sodium measurements, as well as calcium analysis, to fully elucidate the electrophysiological effects of UTs. However, based on current densities and protein levels of I_Kr_, a distinction can already be made between the low and high concentrations of these three UTs.

While there are a large quantity and a variety of UTs present in CKD patients [24], we focused on a subset of PBUTs, including the well-known IS and two metabolites produced during tryptophan metabolism (KYN and KYNA). Since current dialysis regimens are still unable to specifically filter PBUTs, studies such as this one can contribute to the identification of UTs that are more harmful than others, thereby focusing on targets for potential adjustments in blood filtration approaches or adjustments in dietary programs. Recent clinical orientated studies have already shown superior improvements compared to conventional hemodialysis with the use of mixed matrix membranes to specifically filter out PBUTs [25,26].

In this study, we observed a concentration-dependent reduction in I_Kr_ current densities and hERG protein levels as caused by IS and KYNA, while KYN had mild effects. Unfortunately, these differences were not reproducible in either of the hiPSC-CM experiments, in which all three UTs were equally detrimental under all conditions. While it is possible that these three UTs ultimately do not cause different electrophysiological responses, it is important to note that measuring ion currents using patch clamp electrophysiology can be more sensitive to small differences in compound concentrations compared to optical electrophysiological measurements. Unfortunately, patch clamp electrophysiology is also much more labor intensive, restricting applicability of this latter approach for the large-scale reproducibility of these experiments for other UTs in the future.

The allosteric modulator LUF7244 acts as a specific activator of the I_Kr_ current [19,27,28]. Previously, application of LUF7244 resulted in increased I_Kr_ currents, even independent of changes in protein levels [29]. Here, we used LUF7244 as a proof-of-concept method to investigate whether repolarization functionality could be rescued while simultaneously showing the potential of an alternative treatment option for specific electrophysiological effects caused by UTs. LUF7244 completely reversed the APD prolongation caused by all three UTs at their highest concentrations to a similar APD seen in control conditions. This outcome importantly indicates that, regardless of the specific effects of UTs on ion channels other than I_Kr_, the ultimate electrophysiological disturbances can be modulated to stabilize APD repolarization in cardiomyocytes.

## 4. Materials and Methods

### 4.1. Cell Lines

#### 4.1.1. hiPSCs Culture

Commercially obtained ATCC hiPSC cells, originating from a healthy man (ATCC, CS-1026), were cultured daily with Essential 8^TM^ medium (Gibco, A1517001, Waltham, MA, USA), in Geltrex^TM^ LDEV-Free, hESC-Qualified, Reduced Growth Factor Basement Membrane Matrix-coated wells (Gibco, A1413302). The cells were passaged once the confluency reached 80–100%. They were dissociated with TrypLE Express Enzyme (Gibco, 12605010) for 5 min at 37 °C, after which Essential 8^TM^ medium supplemented with 1 µM thiazovivin (Sigma, 420220, St. Louis, MO, USA) was added. The cell suspension was then transferred to a Falcon tube and centrifuged for 3 min at 300 RCF. Finally, the cells were seeded at 15.000 cells/cm^2^ in Essential 8^TM^ medium with 1 µM thiazovivin overnight, after which the medium was refreshed with plain Essential 8^TM^ medium.

#### 4.1.2. hiPSC Cardiomyocyte Differentiation

hiPSCs were cultured until 80–90% confluency and washed with dPBS (Gibco, 14190094). The cells were then provided with RMPI++ (bare RPMI-1640-Medium-GlutaMAX^TM^Supplement-HEPES (Gibco, 72400-021) supplemented with 0.5 mg/mL human recombinant albumin (Sigma, A9731), 0.2 mg/mL L-ascorbic acid 2-phosphate (Sigma, A8960)), and 4 µM CHIR99021 (Sigma, 361559). After 48 h, the medium was replaced with RMPI++ and 5 µM IWP2 (Sigma, 681671) after a single rinse with bare RMPI-1640. Then, the cells were refreshed every other day with RMPI++, for 4 days. Thereafter, the cells were cultured every 3–4 days with bare RMPI, supplemented with B-27^TM^ Supplement (Gibco, 17504001). Prior to the optical experiments, the hiPSC-CMs were dissociated with TrypLE Select Enzyme without phenol red (Gibco, A1217703) and seeded on Geltrex^TM^ coated coverslips.

#### 4.1.3. HEK-hERG Cells

Human embryonic kidney cells stably expressing K_V_11.1 (HEK-hERG, generously obtained from C.T. January [30]) were cultured in Dulbecco’s modified eagle medium, supplemented with 10% fetal calf serum, 1% L-glutamine, and 1% streptomycin/penicillin. The cells were passaged twice per week and cultured at 37 °C and 5% CO_2_.

### 4.2. Drugs and Toxins

The I_Kr_ blocker dofetilide (Sigma, PZ0016), as well as the three UTs—IS (Sigma, I3875), KYN (Sigma, K8625), and KYNA (Sigma, K3375)—were dissolved in DMSO to 10 mM. LUF7244 was custom synthesized at the Division of Drug Discovery and Safety, Leiden, the Netherlands, as described previously [19], and was dissolved in DMSO to 100 mM. All stock solutions were filtered and stored at −20 °C. Clinically relevant total UT concentrations were identified previously [13] and applied in all experiments. Albumin was present in all culture media to establish clinically relevant, free fraction concentrations of all UTs.

### 4.3. Electrophysiology

#### 4.3.1. Optical Action Potential Measurements

hiPSC-CMs were seeded at a density of approximately 150.000 per Geltrex-coated coverslip, which facilitated the formation of cell clusters. Coverslips were incubated with Powerload and FluoVolt (ThermoFischer F10488, 1:1000, Waltham, MA, USA) in complete RMPI medium at 37 °C for 20 min. Action potentials were recorded at 37 °C, during which the coverslips were placed in a bath solution containing (mM): NaCl (130), KCl (4), CaCl_2_ (1.8), MgCl_2_ (1.2), NaHCO_3_ (18), HEPES (10), and glucose (10), with a pH of 7.4. Fluorescent dye signals were recorded on a custom-built microscope (Cairn Research, Faversham, UK) using a 10× objective. Blue light was filtered using a 482/35 excitation filter and projected onto the objective with a 515-nm dichroic mirror. Fluorescent signals were captured via a 514 long-pass emission filter by a high-speed camera (Andor Zyla 5.5.CL3, Oxford Instruments, Abingdon, UK). Analysis of the data was performed using Fiji and Peaks, a custom-written Matlab script (DOI: 10.17605/OSF.IO/86UFE). APD of spontaneously beating hiPSC-CMs was determined at 90% repolarization and then adjusted for the beating rate using a modified Fredericia’s correction: APDcorrected = APD/(∛(60/BPM).

#### 4.3.2. Patch Clamp Electrophysiology

HEK-hERG cells were cultured on poly-L-lysine-coated glass coverslips. Whole-cell voltage clamp recordings were performed at room temperature using an AxoPatch 200B patch clamp amplifier and pClamp software, version 10. Pipettes were pulled using a Sutter P2000 puller, with a resistance of 1.5–3 MΩ. Measurements were performed using an external bath solution containing (mM): NaCl (140), KCl (4), CaCl_2_ (2), MgCl_2_ (1), and HEPES (10) with a pH of 7.4 and an internal pipette solution containing (mM): KCl (110), CaCl_2_ (5.17), MgCl_2_ (1.42), HEPES (10), K2ATP (4), and EGTA (10) with a pH of 7.2. I_Kr_ was measured from a holding potential of −80 mV, followed by a pulse protocol from −60 mV to +60 mV in increments of 10 mV. Currents were analyzed using the pClamp software, with the maximum tail current at each voltage step being depicted in the current-voltage graphs.

### 4.4. Western Immunoblotting

Cultured cells were harvested in RIPA lysis buffer (20 mM Tris, 150 mM NaCl, 10 mM Na_2_HPO4, 1 mM EDTA, 50 mM NaF, 1% Triton X100, 1% Na-deoxycholate, 0.1% SDS, supplemented with 0.2% aprotinin and 1 mM PMSF. SDS-PAGE was performed using 10–20 µg of protein per sample and subsequently transferred onto nitrocellulose membranes. Ponceau staining was used to determine equal loading. hERG primary antibody (Alomone, APC-062, Jerusalem, Israel) and the secondary HRP conjugated antibody (Bio-Rad, 170-6515, Hercules, CA, USA) were diluted in 5% protifar TBST (20 mM Tris, 150 mM NaCl, 0.05% Tween-20) and incubated at 4 °C overnight or at room temperature for 90 min. Data were imaged and analyzed using an ECL detection kit (Cytiva, RPN2232, Marlborough, MA, USA) and a ChemiDoc Molecular Imager and ImageLab software version 6.1. (Bio-Rad).

### 4.5. Quantitative PCR

Total RNA was extracted from HEK-hERG cells using TRIzol reagent (ThermoFischer). After DNAse treatment, DNAse-treated RNA was converted into complement DNA (cDNA) with reverse transcriptase (Invitrogen, Waltham, MA, USA) according to the manufacturer’s protocol. RT-qPCR was performed using TaqMan gene expression assays (Applied Biosystems by Life Technologies Corp., Carlsbad, CA, USA). Relative mRNA levels were determined for *KCNH2* (Hs00542479_g1). TATA-binding protein (Hs00427620_m1), 60S ribosomal protein L32 (Hs04194521_s1) and peptidylprolyl isomerase A (Hs00851655_g1) were used for internal controls. Fold changes were calculated using the 2^(−ΔΔCT)^ method.

### 4.6. Statistical Analysis

All statistical analysis was performed using GraphPad Prism software, version 9. Comparisons were performed using a two-way ANOVA, with Tukey’s post-hoc test to correct for multiple comparisons. Data are reported as individual datapoints (optical electrophysiology), as means ± SEMs (manual electrophysiology), or as means ± SDs (western immunoblotting and quantitative PCR).

## 5. Conclusions

In this study, we have identified the detrimental effects of three UTs on cardiac repolarization. The consequences of clinically relevant concentrations of IS, KYN, and KYNA could be distinguished based on their molecular and functional effects after chronic exposure. Together, these data accentuate the need for further in-dept analysis of the electrophysiological disturbances triggered by UT exposure during CKD. Similar studies in the future could aid in providing a clinically relevant overview of (patho-)physiological concentrations of specific UTs. With this knowledge, research into UTs and potential methods for their removal from the bloodstream can be specified for the most detrimental UTs. Meanwhile, there is validity in modulating APD stability using, e.g., repolarization activators to prevent electrophysiological disturbances caused by UTs.

## Figures and Tables

**Figure 1 ijms-24-05373-f001:**
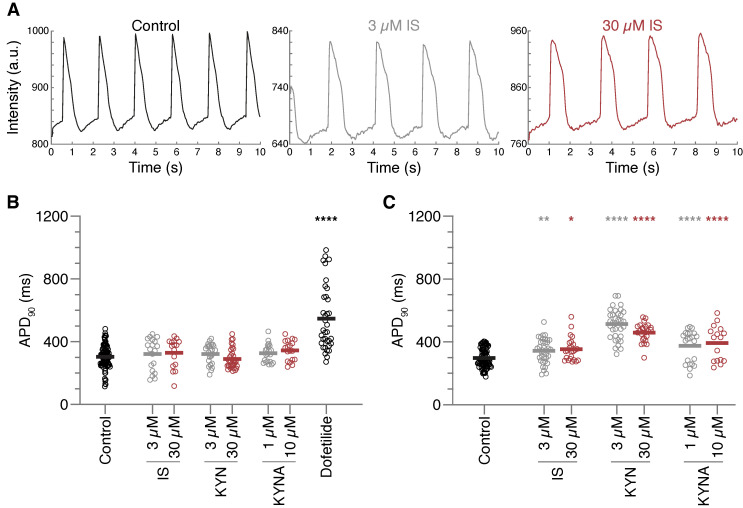
Action potential prolongation induced by chronic uremic toxin exposure. Action potential duration (APD) was measured in spontaneously beating cells, using a voltage sensitive dye, after exposure to two concentrations of indoxyl sulfate (IS), kynurenine (KYN), or kynurenic acid (KYNA). (**A**) Representative action potential traces of chronic exposure measurements in control, 3 µM IS, and 30 µM IS. (**B**) Quantified APD measurements after acute uremic toxin (UT) exposure of 5 min. (**C**) Quantified APD measurements after chronic 48-h UT exposure. Data are shown as means with individual measurements, and quantified APDs were corrected for beating rate. Measurements were obtained in three independent differentiations of hiPSC-CMs, with numbers of cells = 16–99 (**B**), and 16–82 (**C**). * *p* < 0.05, ** *p* < 0.01, **** *p* < 0.0001 compared to control.

**Figure 2 ijms-24-05373-f002:**
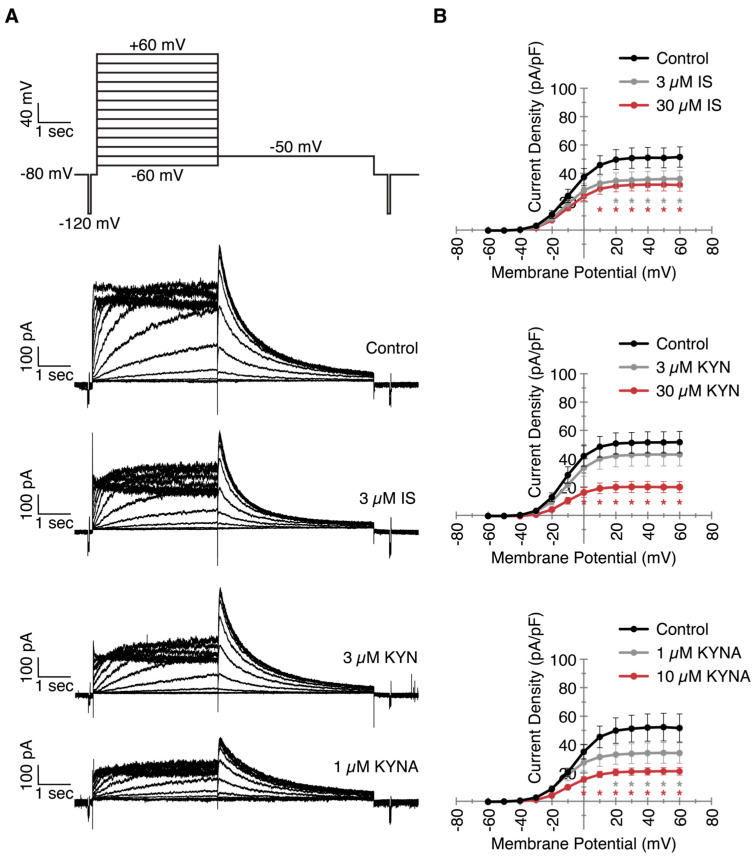
Chronic exposure to uremic toxins decreases I_Kr_ current density. HEK cells stably expressing the hERG ion channel were exposed to two concentrations of indoxyl sulfate (IS), kynurenine (KYN), and kynurenic acid (KYNA) for 48 h. I_Kr_ currents were measured using patch clamp electrophysiology. (**A**) Patch clamp measurement protocol and example traces of control, 3 µM IS, 3 µM KYN, and 1 µM KYNA as clinically relevant concentrations. (**B**) Quantification of all measurements visualized in current-voltage curves. Data are shown as the mean ± SEM. Number of cells = 12–19. * *p* < 0.05 compared to control.

**Figure 3 ijms-24-05373-f003:**
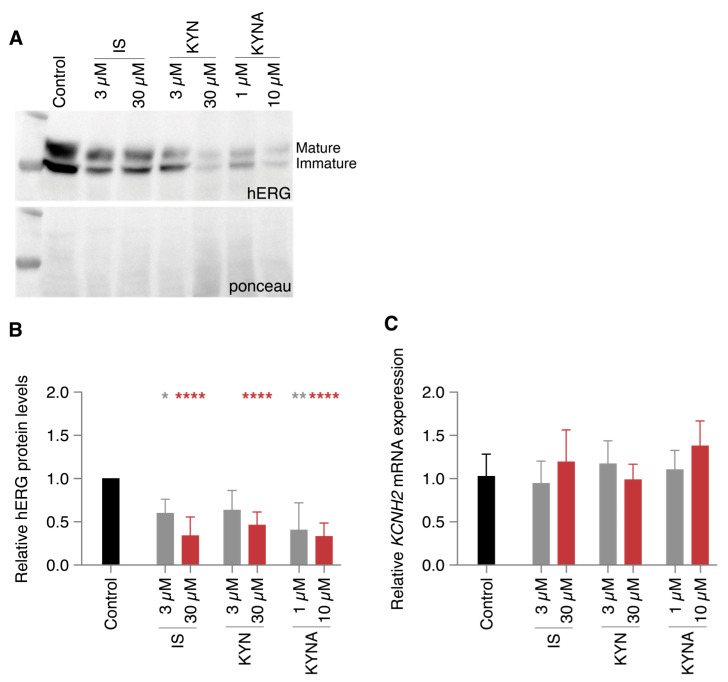
Diminished hERG protein levels caused by chronic uremic toxin exposure. HEK cells stably expressing the hERG ion channel were exposed to two concentrations of indoxyl sulfate (IS), kynurenine (KYN), and kynurenic acid (KYNA) for 48 h. (**A**,**B**) Example and quantification of protein levels. (**C**) Quantification of *KCNH2* mRNA levels. Data are shown as the mean ± SD. Numbers of biological repeats = 3–5 (**B**) and 5 (**C**). * *p* < 0.05, ** *p* < 0.01, **** *p* < 0.0001 compared to control.

**Figure 4 ijms-24-05373-f004:**
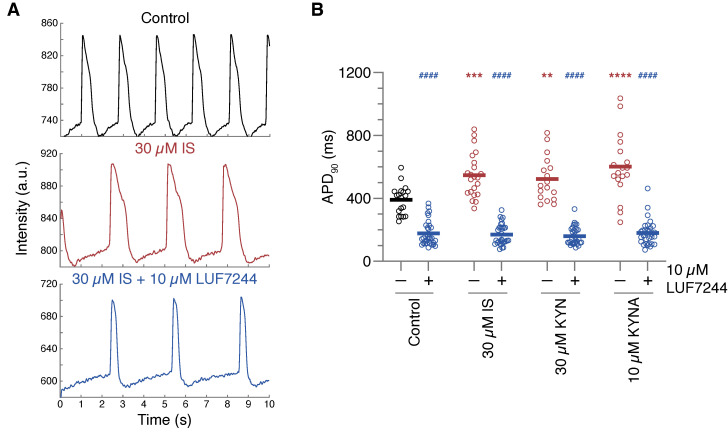
Reversal of uremic toxin-induced action potential prolongation by LUF7244. Action potential duration (APD) was measured in spontaneously beating cells, using a voltage sensitive dye. hiPSC-CMs were exposed to high concentrations of indoxyl sulfate (IS), kynurenine (KYN), and kynurenic acid (KYNA), for 48 h, followed by an additional 24 h of uremic toxin exposure in the absence or presence of LUF7244 (indicated by – and + respectively). (**A**) Example traces of APD measurements in control, 30 µM IS, and 30 µM IS + 10 µM LUF7244. (**B**) Quantification of APD measurements. Data are shown as the means with individual measurements, and quantified APDs were corrected for beating rate. Measurements were obtained in four independent differentiations of hiPSC-CMs with the number of cells = 17–32. ** *p* < 0.01, *** *p* < 0.001, **** *p* < 0.0001 compared to control without LUF7244. #### *p* < 0.0001 compared to control with LUF7244.

## Data Availability

The data presented in this study are available on request from the corresponding author.

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
