# Peer review of "Pro-Arrhythmic Potential of Accumulated Uremic Toxins Is Mediated via Vulnerability of Action Potential Repolarization"

_ijms, 2023, doi:10.3390/ijms24065373_

Round 1
Reviewer 1 Report
The article describes the effects of uremic toxins on repolarization and IKur in vitro. The question addressed is very interesting, mainly due to the lack of removal of uremic toxins during dialysis. Also, the authors explore an alternative solution with a current activator. These results will be useful for further exploring the complex mechanisms involved in sudden cardiac death associated with chronic kidney disease.
Reviewer 2 Report
Chronic kidney disease (CKD) is often associated with cardiac remodelings which could lead to sudden cardiac death. This study used three endogenously produced uremic toxins in hiPSC-CM and HEK cells to investigate if UTs could alter action potential repolarization by specifically investigating hERG channels. They found that UTs resulted in action potential duration (APD) prolongation in hiPSC-CMs independent of UTs concentration. They further figured out in HEK cells that this prolongation was caused by decreased IKr current. Therefore, they concluded that UTs have detrimental effects on cardiac repolarization through changing IKr current.
However, this study only demonstrated that APD prolongation was caused by diminished IKr with these three UTs without providing evidence of how UTs alter hERG channel properties. I would suggest investigating the activation and inactivation kinetics of IKr current.
In the third paragraph of the Introduction section, a brief overview of current studies about cardiac electrophysiological remodeling in CKD is required prior to the proposal of the research question. A logic bridge between SCD and APD prolongation is needed. For example, QT interval prolongation is common in CKD patients and also associated with SCD and APD prolongation, however, the related mechanism of QT interval prolongation is not clearly elucidated. So far, it looks that transient outward potassium current and sodium current are also altered in CKD (PMID: 36595632), which could also contribute to APD prolongation. In addition, I would suggest investigating if UTs change sodium current because incomplete sodium channel inactivation increases late sodium current which also leads to APD prolongation. Revealing the nature of APD prolongation in CKD by investigating multiple ionic channels could help us completely understand how UTs alter cardiac electrophysiology.
In Figure 1, please indicate APD90 was corrected by beats in the annotation. I am curious if paired optical mapping experiments can be done. Could the dye be washed away after optical mapping and then re-introduced after 48-hour culture for another optical mapping?
The purpose of Figure 2 is to demonstrate that IKr was decreased by UTs, since 3-uM KYN didn’t change IKr, I would suggest using representative tracings with high concentrations of UTs. Also regarding 3-uM KYN, this dose has a great effect on APD prolongation in hiPSC-CMs, while didn’t alter IKr in HEK cells, which suggests that KYN may have an ‘off-target’ effect.
If the document with western blot raw data is a supplement, please indicate the places in the manuscript. Otherwise, I think it is not necessary.
In the abstract, ‘The current study aimed to assess the potentially increased pro-arrhythmic risk of pre-identified UTs…’, however, no arrhythmias were observed in the study. To assess the arrhythmic risk, the stimulation with S1S2 and bursting pacing protocols can be delivered to cells. Please carefully describe the aim of the study based on the results.
Round 2
Reviewer 2 Report
Good work!